# OpenReview forum: "Why Has Predicting Downstream Capabilities of Frontier AI Models with Scale Remained Elusive?"
_ICLR.cc/2025/Conference — Submitted to ICLR 2025_

### Official Review · Reviewer_EESV · 2024-11-03

**Soundness:** 2
**Presentation:** 3
**Contribution:** 2
**Rating:** 5
**Confidence:** 4

**Summary:**

The paper presents a case study demonstrating that measures of downstream performance do not correlate strongly with the scale of large language models, unlike measures of the pre-training performance. The study is conducted on multiple-choice question answering benchmarks, evaluated on downstream metrics like accuracy, brier score, etc with the pre-training metric being the probability mass on the correct choice. The key insight is that correlation with scale progressively reduces as the downstream metrics become increasingly non-linear functions of the pre-training metric. The reason is said to be the lack of consideration of the probability masses on the incorrect choices, which is then shown to be strongly correlated with the (predictable) pertaining metric.

**Strengths:**

- Insightful analysis of the degradation of predictability of metrics of interest with model scaling, highlighting the need for other scale-predictable indicators of performance.
- Extensive empirical evaluation spanning a wide range of model families and benchmark datasets.
- Generally, a well-presented paper.

**Weaknesses:**

- The paper seems to draw some obvious conclusions in places, possibly highlighting an overarching drawback with using correlation-based analysis. Correlation measures the degree of *linear* relationship between two variables. Consequently, in the equation block around L357, it is known a priori that if $Corr(\text{Compute}, \log p_\theta^{\text{Vocab}}(\text{Correct Choice}))$ is high, then dropping the $\log$ is bound to reduce correlation statistic.

This highlights another potential oversight in the analysis:
- Takeaway #1 and #3 rightly suggest thinking about scale-predictable metrics. The study highlights what transformations make metrics unpredictable, in particular, from $\log p_\theta^{\text{Vocab}}(\text{Correct Choice}) \to p_\theta^{\text{Choices}}(\text{Correct Choice})$. With the same reasoning, in conjunction with the first point above, it is highly likely that $\log p_\theta^{\text{Choices}}(\text{Correct Choice})$ will be scale-predictable although $p_\theta^{\text{Choices}}(\text{Correct Choice})$ is not.
  - Experiments like those in Figures 3 and 4, but for $\log p_\theta^{\text{Choices}}(\text{Correct Choice})$ will be insightful to study this, and in line with the overall takeaways of the paper. I am willing to update my score depending on the author's response to this point in particular.

Nitpicks:
- Equation (2): should have a $\propto$.
- $N$ is overloaded, in Equation (6) as well as under compute budget calculations.

**Questions:**

- Figure 2 [bottom left]: The peak of the distribution is narrow, and reaches at most 7% samples. It seems unlikely that that area under the distribution curve covers 100% of the samples. What exactly is this graph denoting, if not the per-sample correlation distribution of *all* samples?
  - Moreover, since curves here are most similar to the green illustrative example [top left], the complementary CDF is expected to flatten out to the very right end (close to 1 on the x-axis), but no plots for experiments in the paper seem to have that pattern. Why is that?
- How do these findings relate to the objective mismatch problem: that the pre-training objective does not align with the evaluation objectives?

---

> ### Author Response · Authors · 2024-11-20
> **Response to Reviewer EESV (Part 1)**
>
> Thank you for your detailed review and insightful questions about our methodology. We particularly appreciate your recognition of our paper's extensive empirical evaluation and presentation quality.
>
> For calibration purposes, we’d like to note that the ICLR 2025 rubric differs slightly from previous similar conferences. For example:
>
> - To indicate "Accept", the NeurIPS 2024 rubric says to use 7 whereas the ICLR 2025 rubric says to use 8
> - To indicate "Strong Accept", the NeurIPS 2024 rubric says to use 9 whereas the ICLR 2025 rubric says to use 10
>
> To address the concerns you raised:
>
> > The paper seems to draw some obvious conclusions in places, possibly highlighting an overarching drawback with using correlation-based analysis. Correlation measures the degree of linear relationship between two variables.
>
> We understand this concern, but would like to clarify a factual error. Our analysis employs three distinct correlation metrics:
>
> 1. Pearson
> 2. Spearman
> 3. Kendall
>
> Of these three, only Pearson is linear. Spearman and Kendall are rank correlations and can capture nonlinear relationships. The fact that all three metrics show consistent trends strengthens our conclusions beyond linear relationships.
>
> > Consequently, in the equation block around L357, it is known a priori that [...] then dropping the $\log$ is bound to reduce correlation statistic.
>
> We respectfully disagree with this assertion for two reasons:
>
> 1. Spearman and Kendall correlations are rank correlations. Since $\log$ is a monotonic transformation, Spearman and Kendall correlation scores will be unaffected.
>
> 2. In general, for Pearson correlation, dropping the log can actually increase correlation. Consider correlating $X := Z$ with $Y = \log(Z)$ for random variable $Z$;  the correlation would not decrease when replacing from $Y$ with $Z$ and would probably increase (if $Z$ isn't some unusual distribution)
>
> While this transformation does affect results, we focus on the more consequential transformations that follow it in the pipeline, i.e., interactions with probability mass on incorrect choices.
>
> > Experiments like those in Figures 3 and 4, but for $\log p_{\theta}^{\text{Choices}}(\text{Correct Choice})$ will be insightful to study this, and in line with the overall takeaways of the paper. I am willing to update my score depending on the author's response to this point in particular.
>
> Figure 3 displays distributions of Spearman correlations. Because Spearman is a rank correlation and because $\log$ is a monotonic transformation, the distribution of correlations for $\log p_{\theta}^{\text{Choices}}(\text{Correct Choice})$ will be identical to the distribution of correlations for $p_{\theta}^{\text{Choices}}(\text{Correct Choice})$.
>
> Figure 4 displays Pearson correlations (which we chose to demonstrate how our results hold regardless of which correlational metric one chooses), but Appendix F.2 and Appendix F.3 show Spearman and Kendall correlations, respectively. For the same reason that these are rank correlations and log is a monotonic function, in these figures, $\log p_{\theta}^{\text{Choices}}(\text{Correct Choice})$ will again be identical to $p_{\theta}^{\text{Choices}}(\text{Correct Choice})$.

---

> ### Author Response · Authors · 2024-11-20
> **Response to Reviewer EESV (Part 2)**
>
> > Equation (2): should have a $\propto$
>
> We believe the equation is exact, not proportional. The negative log likelihood for a one-hot target simplifies to:
>
> $$\mathcal{L}_{\theta}^{\text{Vocab}} := - \sum_{v} p^*(V=v|context) \log p_{\theta}(V=v|context)$$
>
> Since the single-token target is a one-hot probability vector, letting $v^*$ denote the token, the negative log likelihood simplifies to:
>
> $$\mathcal{L}_{\theta}^{\text{Vocab}} = - \log p_{\theta}(V=v^*|context)$$
>
> Rearranging, we find Equation 2 is exact:
>
> $$\exp(-\mathcal{L}_{\theta}^{\text{Vocab}}) = p_{\theta}(V=v^*|context)$$
>
> If you disagree or if our exposition can be improved, please let us know.
>
> > $N$ is overloaded, in Equation (6) as well as under compute budget calculations.
>
> Thank you for catching this - we will modify the notation in Equation 6 to use a different variable.
>
> > Figure 2 [bottom left]: The peak of the distribution is narrow, and reaches at most 7% samples. It seems unlikely that that area under the distribution curve covers 100% of the samples. What exactly is this graph denoting, if not the per-sample correlation distribution of all samples?
>
> You've identified an important labeling issue. The y-axis represents a kernel density estimate of the score-compute correlations, not raw sample percentages. We will update the label to clarify this.
>
> > Moreover, since curves here are most similar to the green illustrative example [top left], the complementary CDF is expected to flatten out to the very right end (close to 1 on the x-axis), but no plots for experiments in the paper seem to have that pattern. Why is that?
>
> We’re not sure we follow. Looking at Figure 2, the bottom right subfigure shows the pattern you describe - the complementary CDF remains near 1 until dropping around correlation values close to 1 (>0.75). Could you please clarify?
>
> > How do these findings relate to the objective mismatch problem: that the pre-training objective does not align with the evaluation objectives?
>
> While the objective mismatch problem is important, our work takes a different focus. Rather than questioning the training setup, we investigate why predicting downstream capabilities is difficult given these models as they exist. This allows us to identify specific mechanisms that make prediction challenging, regardless of the training objective.
>
> We'd be happy to expand on any of these points in the revision. In particular, would you like to see additional experiments or analyses that would help address your concerns?

---

> > ### Comment · Reviewer_EESV · 2024-11-21
> >
> > Thanks for the clarification response.
> >
> > > While this transformation does affect results [...] + Figure 4 displays Pearson correlations (which we chose to demonstrate how [...]
> >
> > The figures in Appendix F with other correlation metrics, show a *much* noisier version of the trend conveyed by Figure 4. Since the authors meticulously use various correlation metrics for their analysis, them conveying this difference in the confidence of trends across the correlation metrics in the main text would be insightful. Subsequently, if an assertion is made about the correlation $p_{\theta}^{\text{Choices}}(\text{Correct Choice})$, as in the equation block around [L357], it is best verified empirically.
> >
> > > Looking at Figure 2, the bottom right subfigure shows the pattern you describe - the complementary CDF remains near 1 until dropping around correlation values close to 1 (>0.75). Could you please clarify?
> >
> > Consider the shape of the illustrative blue plot in the top left sub-figure of Figure 2. Since it drops (close to 0) before reaching a correlation of 1 (right-most extreme) the corresponding complementary CDF (top right) flattens out just before reaching 1 on the correlation axis. For a similar drop noticed in all plots of the bottom left figure, a corresponding flattening out is not seen in the bottom left figure (empirical complementary CDF). Why may that be?
> >
> > I thank the authors for clarifications and other comments about updates to the manuscript. For now, I will maintain my ICLR-calibrated score.

---

> > ### Author Response · Authors · 2024-11-24
> > **Corrected Y-Label for Subfigure of Figure 2**
> >
> > > Figure 2 [bottom left]: The peak of the distribution is narrow, and reaches at most 7% samples. It seems unlikely that that area under the distribution curve covers 100% of the samples. What exactly is this graph denoting, if not the per-sample correlation distribution of all samples?
> >
> > As promised, we correct the lower left subfigure of Figure 2's y label to "Estimated Density."

---

> ### Author Response · Authors · 2024-11-24
> **Response to Reviewer EESV Regarding Additional Experiments for Figs 3 & 4**
>
> To quickly recap,
>
> 1. Reviewer EESV requested additional experiments for Figs 3&4 but for $\log p_{\theta}^{Choices}(\text{Correct Choice})$
>
> 2. We answered that Fig 3 already (implicitly) displays that metric because the figure displays $p_{\theta}^{Choices}(\text{Correct Choice})$ and because Spearman is a rank correlation that is not affected by monotonic transformations such as $\log(\cdot)$. We further directed Reviewer EESV to Appendix F.2 and Appendix F.3 for additional visualizations.
>
> 3. Reviewer EESV ignored the first half of our response and wrote, "The figures in Appendix F with other correlation metrics, show a much noisier version of the trend conveyed by Figure 4."
>
> We do not agree. We believe that the figures in Appendices F.2 and F.3 show clear and consistent evidence with Figures 3 & 4. To be more quantitatively precise, we computed what fraction of rows in Figures F.1, F.2, and F.3 exhibit the ordering of metrics that we claim they should in Section 4. We added this paragraph to our paper.
>
> **To quantitatively confirm that the correlation scores indeed follow this ordering, we computed what fraction of (benchmark, correlation metric, model family, correlation distribution statistic) tuples obey the ordering. To be maximally conservative, we checked for strict inequalities only. We found that across benchmarks, model families, and the 4 correlation distribution statistics, the claimed ordering of metrics held at least 82.4\% of the time for Pearson, 85.6\% for Spearman and 90.4\% for Kendall.**
>
> Hopefully this analysis provides stronger evidence and addresses Reviewer EESV's concerns.

---

### Official Review · Reviewer_QwKS · 2024-11-03

**Soundness:** 4
**Presentation:** 3
**Contribution:** 3
**Rating:** 6
**Confidence:** 3

**Summary:**

This paper investigates why predicting downstream performance of large language models has remained challenging, despite the relative predictability of pretraining loss. The authors analyze multiple model families across various benchmarks and identify one factor which reduces the predictability of benchmark performance: people tend to focus on scaling laws looking at benchmark accuracy scores, rather than further upstream metrics such as the model logprobs on the correct choice (without renormalization to valid options).

**Strengths:**

The main strength of the paper is that the story is quite simple and is strongly supported by the experiments. It is also timely and a matter that has many downstream implications.

The empirical evaluation seems to be quite comprehensive and technically sound.

The authors also provide both a compelling mechanistic explanation as to why the probability mass on the incorrect choices matter, and actionable insights for the field moving forward in light of these results.

**Weaknesses:**

While the findings are interesting, I'm a bit confused as to why the authors did not try to use their insights to attempt novel predictions of performance on benchmarks, and just limited themselves to measuring correlations.

Also, I think I'm a bit confused about one of the takeaways: they say to use p^Vocab, but AFAICT in section 6 they also argue that to do even better in predicting benchmark scores, one would need to model the joint evolution of probabilities across all tokens. Do the authors think that the evolution of probabilities across other tokens is a large contributing factor to having good scaling laws? From the results from Figure 3, it seems like we already have very large correlations when using log p^Vocab?

To broadly recap, I think most of my confusion stems from the disconnect between the promising analysis using correlations, and the lack of actual predictions using their best technique (log p^Vocab), showing how they fit the performance that is actually obtained by models. It may be that I'm misunderstanding something.

**Questions:**

"All the scores we discuss are per-datum." / "Firstly, this negative log likelihood is not computed in expectation over a corpus;" -> What does this mean? What would it even look like to do the log likelihood in expectation over the corpus?

Is it fair to say that one of the main takeaways of your paper is that for best results, you would need to "model the joint evolution of probabilities across all tokens, not just the correct value" (and this is what you do in Section 6)?

You say "one must predict not just how probability mass concentrates on correct choices with scale, but also how probability mass fluctuates on incorrect choices with scale." -> would it be more fair to say that you think that predicting how probability mass fluctuates on incorrect choices *may* enable better predictions, but you don't really know by how much? Unless I'm missing something, you don't have strong evidence that it would improve things substantially? The results from Figure 3 seem already quite strong without this additional modeling?

---

> ### Author Response · Authors · 2024-11-20
> **Response to Reviewer QwKS**
>
> Thank you for your thoughtful review. We particularly appreciate your recognition of our paper's strong empirical support and mechanistic explanations.
>
> For calibration purposes, we’d like to note that the ICLR 2025 rubric differs slightly from previous similar conferences. For example:
>
> - To indicate "Accept", the NeurIPS 2024 rubric says to use 7 whereas the ICLR 2025 rubric says to use 8
> - To indicate "Strong Accept", the NeurIPS 2024 rubric says to use 9 whereas the ICLR 2025 rubric says to use 10
>
> To address the concerns you raised:
>
> > While the findings are interesting, I'm a bit confused as to why the authors did not try to use their insights to attempt novel predictions of performance on benchmarks, and just limited themselves to measuring correlations.
>
> > To broadly recap, I think most of my confusion stems from the disconnect between the promising analysis using correlations, and the lack of actual predictions using their best technique (log p^Vocab), showing how they fit the performance that is actually obtained by models. It may be that I'm misunderstanding something.
>
> This is a fair point. Our paper focuses on explaining why predicting downstream capabilities on multiple-choice question-answering (MCQA) benchmarks has remained challenging since GPT-3. We believed it was important to first establish a thorough understanding of the underlying challenges before proposing solutions, because the core science of predictable evaluations is at present underdeveloped. We hope to build groundwork for diverse prediction approaches of both MCQA and other extended evaluation methodologies by making concrete the barriers to prediction in a readily-analyzable case.
>
> > Also, I think I'm a bit confused about one of the takeaways: they say to use p^Vocab, but AFAICT in section 6 they also argue that to do even better in predicting benchmark scores, one would need to model the joint evolution of probabilities across all tokens. Do the authors think that the evolution of probabilities across other tokens is a large contributing factor to having good scaling laws?
>
> > Is it fair to say that one of the main takeaways of your paper is that for best results, you would need to "model the joint evolution of probabilities across all tokens, not just the correct value" (and this is what you do in Section 6)?
>
> Thank you for helping us identify this unclear messaging. We actually make two separate points that we should distinguish more clearly in the revision:
>
> 1. For practitioners seeking scaling-predictable signals: p^Vocab provides strong correlations with compute and can be used directly.
>
> 2. For practitioners specifically needing Accuracy or Brier Score predictions: achieving high-quality predictions may require modeling probability mass on all available choices, including incorrect choices.
>
> We are agnostic to the choice of metric. We aim to help practitioners understand the tradeoffs between different metrics rather than prescribe a single approach. We will revise the text to make this distinction clearer.
>
> > "All the scores we discuss are per-datum." / "Firstly, this negative log likelihood is not computed in expectation over a corpus;" -> What does this mean? What would it even look like to do the log likelihood in expectation over the corpus?
>
> (Negative) log likelihood is the “standard” expected loss that is studied in many research papers, for instance, in the neural scaling literature. More specifically, to measure how performant a model is, researchers compute the expected loss over a corpus by:
>
> 1. Taking $n$ sequences of length $t$
> 2. Computing the negative log likelihood $-\log p_{\theta}(\text{correct token}_{n,t} | \text{previous tokens}_{n, t})$ for each sequence at each index
> 3. Averaging over these $nt$ values
>
> Our analysis differs by examining individual benchmark samples without averaging. This granular approach helps us understand how predictability degrades for specific instances.
>
> > You say "one must predict not just how probability mass concentrates on correct choices with scale, but also how probability mass fluctuates on incorrect choices with scale." -> would it be more fair to say that you think that predicting how probability mass fluctuates on incorrect choices may enable better predictions, but you don't really know by how much?
>
> You make a valid point. Our current language overstates the necessity of modeling incorrect choices. A more accurate statement would be: "Our analysis suggests that modeling probability mass fluctuations on incorrect choices could improve predictions of metrics like Accuracy and Brier Score, though the magnitude of improvement remains an open question for future work." We will revise the manuscript to better reflect this uncertainty while maintaining our key insights about the role of incorrect choices in predictability.

---

> ### Comment · Reviewer_QwKS · 2024-11-22
> **Response**
>
> >For practitioners seeking scaling-predictable signals: p^Vocab provides strong correlations with compute and can be used directly.
> For practitioners specifically needing Accuracy or Brier Score predictions: achieving high-quality predictions may require modeling probability mass on all available choices, including incorrect choices.
>
> Yes, I think emphasizing this would improve the clarity of the submission. Could you provide context for why practitioners would specifically _need_ Accuracy or Brier Score? Is this for cases in which they wouldn't have access to the full probabilities?
>
> ---
>
> Overall, I still stand by my score that this paper is marginally above the acceptance threshold. While I agree that it is "important to first establish a thorough understanding of the underlying challenges before proposing solutions", it's hard to assess the importance of these findings without seeing how much predictions of benchmark scores are improved through them. The paper seems a bit incomplete without them.
>
> Ultimately, I am left with the feeling that the paper spent perhaps too much effort in doing very careful analysis of the phenomenon, when some of that effort could have been better spent trying to apply the findings in the simplest way possible to better demonstrate the importance of the results. I'm not sure if I'm underestimating the amount of work that it would have required to apply the findings with p^Vocab to actual benchmark predictions, but it seems like it shouldn't have been that much additional effort?

---

> ### Author Response · Authors · 2024-11-24
> **Response to Reviewer QwKS**
>
> > Ultimately, I am left with the feeling that the paper spent perhaps too much effort in doing very careful analysis of the phenomenon, when some of that effort could have been better spent trying to apply the findings in the simplest way possible to better demonstrate the importance of the results.
>
> To contrast your feeling that our work was too detailed, Reviewer EESV feels that our analysis was not sufficiently detailed in that (1) we should have additionally studied $\log p_{\theta}^{\text{Choices}}(\text{Correct Choice})$ and (2) the data shown in our Figure 4 should be quantified to make the claim more believable. We will add these analysis for Reviewer EESV soon, but in the interim, **we highlight this tension to point out that our reviewers are simultaneously criticizing the manuscript as excessively detailed and insufficiently detailed; this is a no-win scenario for us**
>
> > Could you provide context for why practitioners would specifically need Accuracy or Brier Score? Is this for cases in which they wouldn't have access to the full probabilities?
>
> - **Accuracy** is a widely used metric that humans generally find interpretable: what fraction of the time does the model "do the right thing?"
>
> - **Brier Score** is a "strictly proper scoring rule" that is more continuous than Accuracy and has been used in large benchmarks, e.g., Google's BIG Bench.
>
> To reiterate, different practitioners have different needs, and our manuscript aims to help illuminate what to expect from choosing one metric over another. We do not advocate one metric over another.
>
> > You say "one must predict not just how probability mass concentrates on correct choices with scale, but also how probability mass fluctuates on incorrect choices with scale." -> would it be more fair to say that you think that predicting how probability mass fluctuates on incorrect choices may enable better predictions, but you don't really know by how much?
>
>
> > You make a valid point. Our current language overstates the necessity of modeling incorrect choices. A more accurate statement would be: "Our analysis suggests that modeling probability mass fluctuations on incorrect choices could improve predictions of metrics like Accuracy and Brier Score, though the magnitude of improvement remains an open question for future work." We will revise the manuscript to better reflect this uncertainty while maintaining our key insights about the role of incorrect choices in predictability.
>
> As promised, we updated the manuscript to be more specific and tempered.

---

> > ### Comment · Reviewer_QwKS · 2024-11-24
> >
> > > we highlight this tension to point out that our reviewers are simultaneously criticizing the manuscript as excessively detailed and insufficiently detailed; this is a no-win scenario for us
> >
> > While I acknowledge this tension, I don't think it is unresolvable. Showing that your findings can lead to better predictions of benchmark scores would have likely helped with addressing the criticisms of Reviewer EESV too: assuming you would have positive results on that, it would be undeniable that – regardless of the remaining limitations of the analysis discussed by Reviewer EESV – your findings have practical value, making such limitations of the analysis less important. Moreover, having those results could likely have guided and informed the analysis on things that seem to make the most difference to the prediction capabilities, which is ultimately what this paper tries to be about (as in the title).
> >
> > In that scenario, I would have been very willing to fight for acceptance of the paper. However, as things stand, I find it hard to be especially excited about the results, while recognizing them as non-trivial and interesting (again, bringing the paper marginally above the acceptance threshold).

---

### Official Review · Reviewer_8JjP · 2024-11-04

**Soundness:** 3
**Presentation:** 3
**Contribution:** 3
**Rating:** 6
**Confidence:** 4

**Summary:**

This paper studies the reasons behind why predicting downstream capabilities of large language models has remained challenging despite well-established scaling laws for pretraining performance. The authors show that the process of computing downstream performance metrics like accuracy from pretraining log likelihoods involves a sequence of transformations that progressively degrade the statistical relationship between performance and scale. They identify the key mechanism causing this degradation: downstream metrics depend not just on the probability mass assigned to the correct answer choice, but also on how probability mass fluctuates on the specific incorrect answer choices. The probability assigned to incorrect choices does not have a strong predictable relationship with scale. The authors argue this explains the comparative unpredictability of downstream performance and suggest paths forward, like modeling the scaling of probability on incorrect choices. They also advise on designing more predictable downstream evaluations tightly coupled to pretraining log likelihoods. Overall, the paper provides valuable insights into the factors affecting downstream predictability and guidance on improving evaluation of frontier models.

**Strengths:**

The key insight that downstream unpredictability is caused by probability fluctuations on incorrect answer choices is novel and not obvious a priori. Framing the problem in terms of a sequence of transformations that degrade predictability is useful to make modular progress on the problem. The experiments are comprehensive, covering many models, benchmarks, performance metrics, and correlation measures. the authorsalso perform robustness checks to establish their claims.

A key strength of the paper is the precise mathematical formulation of how downstream performance metrics are computed from pretraining log likelihoods. This formalism allows the authors to reason about how each step impacts the relationship between performance and scale.
The paper's experimental methodology is also comprehensive. The authors evaluate five model families on twelve diverse multiple-choice benchmarks, covering commonsense reasoning, science, math, social science, and the humanities. For each benchmark, they compute per-sample scores for three performance metrics (accuracy, Brier score, probability on correct choice) across many model scales.

Beyond simply demonstrating that downstream metrics become less correlated with scale after a sequence of transformations, the authors provide a clear mechanistic explanation for why this occurs. They show in Section 5 that that fluctuations in the probability assigned to specific incorrect answer choices breaks the clean relationship between downstream performance and scale.

**Weaknesses:**

The paper focuses exclusively on multiple-choice question-answering benchmarks. While the authors justify this focus, multiple-choice has limitations as an evaluation format. It would strengthen the paper to discuss how the insights might generalize to other types of benchmarks like free-form language generation. Are there analogous factors that could cause unpredictability in other settings?

The experiments focus on predicting performance for individual samples. In practice, aggregate performance over a distribution is often of interest. The authors should also discuss if fluctuations on incorrect choices "average out" over a distribution, and if there are still challenges predicting aggregate performance.
The guidance on designing more predictable evaluations, is very limited. The authors should expand on this point with more details and examples of predictable benchmark designs.

**Questions:**

Some key questions that the authors can explore are as follows:

How can their results and insights about probability fluctuations on incorrect choices generalize beyond multiple-choice benchmarks to other types of language tasks? Are there similar factors that cause unpredictability for other task formats?

The experiments focus on individual samples. Do the challenges with predicting performance due to probability fluctuations persist even when averaging over a distribution of samples? Or do the fluctuations tend to "average out"?

The authors discuss modeling the scaling trends for probability assigned to incorrect answer choices as a path towards better downstream predictability. How feasible do they think this approach is in practice? What challenges do they foresee? How does it scale?

---

> ### Author Response · Authors · 2024-11-20
> **Response to Reviewer 8JjP**
>
> Thank you for your thorough and insightful review. We particularly appreciate your recognition of our paper's mathematical formalism and comprehensive experimental methodology.
>
> For calibration purposes, we’d like to note that the ICLR 2025 rubric differs slightly from previous similar conferences. For example:
>
> - To indicate "Accept", the NeurIPS 2024 rubric says to use 7 whereas the ICLR 2025 rubric says to use 8
> - To indicate "Strong Accept", the NeurIPS 2024 rubric says to use 9 whereas the ICLR 2025 rubric says to use 10
>
> To address the concerns you raised:
>
> > The paper focuses exclusively on multiple-choice question-answering benchmarks.[…] multiple-choice has limitations as an evaluation format. It would strengthen the paper to discuss how the insights might generalize to other types of benchmarks like free-form language generation. Are there analogous factors that could cause unpredictability in other settings?
>
> > How can their results and insights about probability fluctuations on incorrect choices generalize beyond multiple-choice benchmarks to other types of language tasks? Are there similar factors that cause unpredictability for other task formats?
>
> > The guidance on designing more predictable evaluations, is very limited.
>
> This is an excellent point which we've attempted to stress in our paper.
>
> A major conclusion of our paper is the finding that evaluation metrics with "more indirection" via transformations away from the pretraining negative log likelihood (NLL) loss on the correct answer string, including the incorporation of additional information beyond correct-answer NLL, are harder to predict for MCQA.
>
> We believe this general intuition will have a bearing on more complex generative, chain-of-thought, or agentic tasks: the "simpler" the evaluation metric is, and "closer" to more-predictable primitives such as the loss, the more easily predictable it becomes. We hope that future work may attempt to directly design such proxy metrics for generative evaluations.
>
> In our work, we demonstrate why direct prediction of Accuracy is challenging and trace this to the sequence of transformations from negative log likelihoods. We see exciting future directions for scaling-predictable evaluations concerning generative evaluations (e.g., math, coding) and agentic evaluations. The exact details will differ, but the overarching approach will remain the same.
>
> > In practice, aggregate performance over a distribution is often of interest. The authors should also discuss if fluctuations on incorrect choices "average out" over a distribution, and if there are still challenges predicting aggregate performance.
>
> We fully agree that aggregate performance is very often of interest. This question leads to an important insight we will add to the paper: In MCQA, probability mass fluctuations on incorrect choices do not "average out" in the way one might expect. Unlike estimating the mean of a random variable (where positive and negative deviations cancel), the nonlinear nature of metrics like Accuracy and Brier Score means that probability mass shifts between incorrect options affect scores in complex ways that persist under averaging. For example, if probability mass shifts from incorrect option A to incorrect option B, the impact on accuracy depends on whether either option had enough mass to compete with the correct answer - there's no natural cancellation. We could add a paragraph describing this phenomenon to better explain why aggregate performance remains challenging to predict, if you think it would be valuable?

---

> ### Author Response · Authors · 2024-11-24
> **Response to Reviewer 8JjP**
>
> > In practice, aggregate performance over a distribution is often of interest. The authors should also discuss if fluctuations on incorrect choices "average out" over a distribution, and if there are still challenges predicting aggregate performance.
>
> > We fully agree that aggregate performance is very often of interest. This question leads to an important insight we will add to the paper: In MCQA, probability mass fluctuations on incorrect choices do not "average out" in the way one might expect. Unlike estimating the mean of a random variable (where positive and negative deviations cancel), the nonlinear nature of metrics like Accuracy and Brier Score means that probability mass shifts between incorrect options affect scores in complex ways that persist under averaging. For example, if probability mass shifts from incorrect option A to incorrect option B, the impact on accuracy depends on whether either option had enough mass to compete with the correct answer - there's no natural cancellation. We could add a paragraph describing this phenomenon to better explain why aggregate performance remains challenging to predict, if you think it would be valuable?
>
> As promised, we have updated the manuscript adding a paragraph pointing out that errors do not cancel out in MCQA in a "standard" sense.

---

> ### Author Response · Authors · 2024-11-24
> **Requesting Response from Reviewer 8JjP?**
>
> Please let us know if you have any remaining questions or concerns. Thank you!

---

### Official Review · Reviewer_cN5E · 2024-11-04

**Soundness:** 3
**Presentation:** 4
**Contribution:** 3
**Rating:** 6
**Confidence:** 3

**Summary:**

The authors show that down-stream performance for multi-choice question answering can be predicted using alternative surrogates. This has a lot of importance when predicting real world model performance than simply looking at the regression loss.

**Strengths:**

- The authors suggest alternate metrics to predict multi-choice question-answering performance than the current ones that is more predictive in real world benchmarks
- Their alternate metrics are also general (for instance, they don't have conditions on performance being above some level) and easy to compute
- This work should have a good impact on investigating alternate metrics for down-stream performance when pretraining LLMs

**Weaknesses:**

- This work only focuses on multi-choice question answering, although quite general, it still is lacking when we consider even universal methods such as CoT does not directly fall into it, not to mention tool-usage etc.

**Questions:**

- Do you think these ideas will transfer to even larger models? I understand checkpoints for newer and larger models are usually not public.

---

> ### Author Response · Authors · 2024-11-20
> **Response to Reviewer cN5E**
>
> Thank you for your thoughtful review and for highlighting the importance of our work for predicting real-world model performance. We appreciate your recognition of our metrics' generality and ease of computation.
>
> For calibration purposes, we’d like to note that the ICLR 2025 rubric differs slightly from previous similar conferences. For example:
>
> - To indicate "Accept", the NeurIPS 2024 rubric says to use 7 whereas the ICLR 2025 rubric says to use 8
> - To indicate "Strong Accept", the NeurIPS 2024 rubric says to use 9 whereas the ICLR 2025 rubric says to use 10
>
> To address the concerns you raised:
>
> > Do you think these ideas will transfer to even larger models? I understand checkpoints for newer and larger models are usually not public.
>
> This is an excellent question. While we share your desire to validate our findings on larger models, we believe our insights will generalize to more capable models for two fundamental reasons:
>
> 1. Our analysis depends only on how downstream metrics are computed from negative log likelihoods - a relationship that remains constant regardless of model scale
>
> 2. The underlying assumption that better models assign higher probability to valid and plausible tokens is much weaker than assuming power law scaling, and should hold across scales.
>
> > This work only focuses on multi-choice question answering, although quite general, it still is lacking when we consider even universal methods such as CoT does not directly fall into it, not to mention tool-usage etc.
>
> We appreciate this observation and have two responses:
>
> 1. We agree that multiple-choice question answering (MCQA) represents only part of the broader challenge of scaling-predictable evaluations. However, the field has yet to achieve reliable predictions even for MCQA benchmarks, let alone more complex capabilities. We believe establishing strong foundations in the prediction of MCQA benchmark behavior is a crucial first step toward understanding more sophisticated behaviors like Chain-of-Thought (CoT) reasoning or tool use. Notably, our work on quantifying the predictability of different downstream metrics indicates that metrics which intuitively have “higher complexity” are less predictable for MCQA in our work–we believe it is likely that other metrics used for scoring CoT evaluations like extractive exact match have a relatively greater “complexity” for prediction, and this cements our desire to thoroughly investigate the testbed of MCQA to provide a foothold for the study of more complex evaluation approaches such as CoT.
>
> 2. While perhaps less attention-grabbing than tool use or agentic behaviors, MCQA remains highly valuable for practical scaling research: (i) the vast majority of current benchmarks rely on MCQA, (ii) It provides an efficient format for testing diverse capabilities across domains like mathematics, science, and humanities, (iii) it offers clear evaluation criteria make it ideal for studying fundamental scaling relationships, (iv) the insights gained from MCQA can inform hypotheses for predicting more complex capabilities. We would especially like to highlight that more difficult tasks such as involved CoT reasoning or tool-usage are far more difficult for pretrained models, which presents challenges for predicting the scaling behavior of pretrained models as we (and other practitioners served by the use of scaling laws) consider.
> In the revised version, we will better emphasize how our work on MCQA lays groundwork for future research on scaling-predictable evaluations of more advanced capabilities.

---

> ### Author Response · Authors · 2024-11-24
> **Requesting Response from Reviewer cN5E?**
>
> Please let us know if you have any remaining questions or concerns. Thank you!

---

> > ### Comment · Reviewer_cN5E · 2024-11-25
> >
> > Thank you for the detailed rebuttal. After reading your rebuttal and other reviewers' reviews, I would like to keep my score as is (6).

---

### Meta-Review · Area_Chair_51ZR · 2024-12-20

**Metareview:**

The paper presents a case study showing that downstream performance metrics do not strongly correlate with the scale of large language models, in contrast to pre-training performance metrics. The study evaluates downstream metrics, such as accuracy and Brier scores, on multiple-choice question-answering benchmarks, with pre-training metrics being the quality of probability assigned to correct choices.

The analysis perspective is intriguing, as the authors attempt a novel examination of the benchmarks, but they limited their scope to measuring correlations. Moreover, some of the conclusions drawn in the paper are rather unremarkable in the current research context.

**Additional Comments On Reviewer Discussion:**

To ensure an effective decision, the AC considered the comments from all reviewers and thoroughly read the manuscript. Significant concerns remain about the paper, particularly regarding its novelty. Additionally, after a careful review, the AC found the paper to be poorly written, with experimental setups and descriptions often vague or inaccurate, a point also raised by the reviewers. Given the paper’s scores, which are clearly below the acceptance threshold, the decision is to reject the paper.

---

### Decision · Program_Chairs · 2025-01-22

Reject